# DIFFERENTIABLE RENDERING WITH REPARAMETERIZED VOLUME SAMPLING

## ABSTRACT

We propose an alternative rendering algorithm for neural radiance fields based on importance sampling. In view synthesis, a neural radiance field approximates underlying density and radiance fields based on a sparse set of scene views. To generate a pixel of a novel view, it marches a ray through the pixel and computes a weighted sum of radiance emitted from a dense set of ray points. This rendering algorithm is fully differentiable and facilitates gradient-based optimization of the fields. However, in practice, only a tiny opaque portion of the ray contributes most of the radiance to the sum. Therefore, we can avoid computing radiance in the rest part. In this work, we use importance sampling to pick non-transparent points on the ray. Specifically, we generate samples according to the probability distribution induced by the density field. Our main contribution is the reparameterization of the sampling algorithm. It allows end-to-end learning with gradient descent as in the original rendering algorithm. With our approach, we can optimize a neural radiance field with just a few radiance field evaluations per ray. As a result, we alleviate the costs associated with the color component of the neural radiance field at the additional cost of the density sampling algorithm.

## 1 INTRODUCTION

We propose a volume rendering algorithm for learning 3D scenes and generating novel views. Recently, learning-based approaches led to significant progress in this area. As an early instance, (20) represent a scene via a density field and a radiance (color) field parameterized with an MLP. Using a differentiable volume rendering algorithm (18) with the MLP-based fields to produce images, they minimize the discrepancy between the output images and a set of reference images to learn a scene representation. The algorithm we propose is a drop-in replacement for the volume rendering algorithm used in NeRF (20) and follow-ups.

The underlying model in NeRF generates an image point in the following way. It casts a ray from a camera through the point and defines the point color as a weighted sum along the ray. The sum aggregates the radiance of each ray point with weights induced by the density field. Each term involves a costly neural network query, and model has a trade-off between rendering quality and computational load. NeRF obtained a better trade off with a two-stage sampling algorithm obtaining ray points with higher weights. The algorithm is reminiscent of importance sampling, yet it requires training an auxiliary model.

In this work we propose a rendering algorithm based on importance sampling. Our algorithm also acts in two stages. In the first stage, it marches through the ray to estimate density. In the second stage, it constructs a Monte-Carlo color approximation using the density to pick points along the ray. Figure 1 illustrates the estimates for a varying number of samples. The resulting estimate is fully-differentiable and does not require any auxiliary models. Besides, we only need a few samples to construct a precise color approximation. Intuitively, we only need to compute the radiance of the point where a ray hits a solid surface. As a result, our algorithm is especially suitable for recent architectures (23; 36; 32) that use distinct models to parameterize radiance and density. Specifically, the first stage only queries the density field, whereas the second stage only queries the radiance field. Compared to the standard rendering algorithm, the second stage of our algorithm avoids redundant radiance queries and reduces the memory required for rendering at the cost of slight estimate variance.

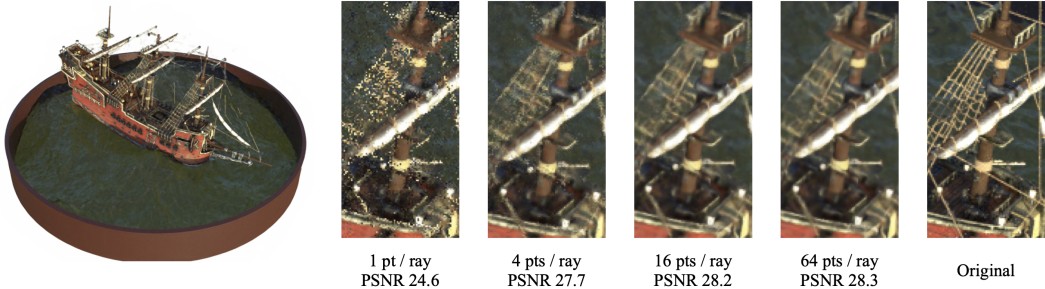

|  1 pt / ray | 4 pts / ray | 16 pts / ray | 64 pts / ray | Original |
| PSNR 24.6 | PSNR 27.7 | PSNR 28.2 | PSNR 28.3 | |

Figure 1: Novel views of a ship generated with the proposed radiance estimates. For each ray we estimate density and then compute radiance at a few ray points generated using the ray density. As the above images indicate, render quality gradually improves with the number of ray points and saturates at approximately 16 ray points.

Below, Section 2 give a recap of neural radiance fields. Then we proceed to the main contribution of our work in Section 3, namely the rendering algorithm fueled by a novel sampling procedure. Finally, in our experiments in Section 5 we evaluate the algorithm in terms of rendering quality, speed and memory requirements.

## 2 NEURAL RADIANCE FIELDS

Neural radiance fields represent 3D scenes with a non-negative scalar density field $\sigma : \mathbb{R}^3 \to \mathbb{R}^+$ and a vector radiance field $c : \mathbb{R}^3 \times \mathbb{R}^3 \to \mathbb{R}^3$. Scalar field $\sigma$ represents volume density at each spatial location $\boldsymbol{x}$, and $c(\boldsymbol{x}, \boldsymbol{d})$ returns the light emitted from spatial location $\boldsymbol{x}$ in direction $\boldsymbol{d}$ represented as a normalized three dimensional vector.

For novel view synthesis, NeRF adapts a volume rendering algorithm that computes pixel color $C(\boldsymbol{r})$ (denoted with a capital letter) as expected radiance for a ray $\boldsymbol{r} = \boldsymbol{o} + t\boldsymbol{d}$ passing through a pixel from origin $\mathbf{o} \in \mathbb{R}^3$ in direction $\mathbf{d} \in \mathbb{R}^3$. To ease the notation, we will denote density and radiance restricted to ray $\boldsymbol{r}$ as

$$\sigma_{\boldsymbol{r}}(t) := \sigma(\boldsymbol{o} + t\boldsymbol{d}) \tag{1}$$
$$c_{\boldsymbol{r}}(t) := c(\boldsymbol{o} + t\boldsymbol{d}, \boldsymbol{d}). \tag{2}$$

With that in mind, the expected radiance along ray $\boldsymbol{r}$ is given as

$$C(\boldsymbol{r}) = \int_{t_n}^{t_f} p_{\boldsymbol{r}}(t) c_{\boldsymbol{r}}(t) \mathrm{d}t, \text{ where } p_{\boldsymbol{r}}(t) := \sigma_{\boldsymbol{r}}(t) \exp\left(-\int_{t_n}^{t} \sigma_{\boldsymbol{r}}(s) \mathrm{d}s\right). \tag{3}$$

Here, $t_n$ and $t_f$ are *near* and *far* ray boundaries and $p_{\boldsymbol{r}}(t)$ is an unnormalized probability density function of a random variable t on a ray $\boldsymbol{r}$. Intuitively, t is the location on the ray where a portion of light coming into the point $\boldsymbol{o}$ was emitted.

To approximate the nested integrals in Equation 3, Max (18) proposed to replace fields $\sigma_{\boldsymbol{r}}$ and $c_{\boldsymbol{r}}$ with a piecewise approximation on a grid $t_n = t_0 < t_1 < \cdots < t_m = t_f$ and compute the formula 3 analytically for the approximation. In particular, a piecewise constant approximation, which is predominant in NeRF literature, yields formula

$$\hat{C}(\boldsymbol{r}) = \sum_{i=1}^{m} (1 - \exp(-\sigma_{\boldsymbol{r}}(t_i)\delta_i)) \exp\left(-\sum_{j=1}^{i-1} \sigma_{\boldsymbol{r}}(t_j)\delta_j\right) c(t_i), \text{ where } \delta_i := t_{i+1} - t_i. \tag{4}$$

Importantly, Equation 4 is fully differentiable and can be used as a part of gradient-based learning pipeline.

Given the ground truth expected color $C_{gt}(\boldsymbol{r})$ along $\boldsymbol{r}$, the optimization objective in NeRF

$$L(\hat{C}(\boldsymbol{r}), C_{gt}(\boldsymbol{r})) = \|\hat{C}(\boldsymbol{r}) - C_{gt}(\boldsymbol{r})\|_2^2 \tag{5}$$

captures the difference between $C_{gt}(\boldsymbol{r})$ and estimated color $\hat{C}(\boldsymbol{r})$. To reconstruct a scene NeRF runs a gradient based optimizer to minimize the objective 5 averaged across multiple rays and multiple viewpoints.

While the above approximation works in practice, it involves multiple evaluations of $c$ and $\sigma$ along a dense grid. Besides that, a common situation is when a ray intersects a solid surface at some point $t \in [t_n, t_f]$. In this case, probability density $p_{\boldsymbol{r}}(t)$ will concentrate its mass near $t$ and will be close to zero in other parts of the ray. As a result, most of the terms in Equation 4 will make negligible contribution to the sum. In Section 4, we discuss various solutions to picking the grid points that are most likely to contribute to the sum. As an alternative, in the next section we propose to estimate the expected radiance with stochastic estimates that require only few radiance evaluations.

## 3 STOCHASTIC ESTIMATES FOR THE EXPECTED COLOR

Monte Carlo method gives a natural way to approximate the expected color. For example, given $k$ i.i.d. samples $t_1, \ldots, t_k \sim p_{\boldsymbol{r}}(t)$ and the normalization constant $y_f := \int_{t_n}^{t_f} p_{\boldsymbol{r}}(t)\mathrm{d}t$, the following sum

$$\hat{C}_{MC}(\boldsymbol{r}) = \frac{y_f}{k} \sum_{i=1}^{k} c_{\boldsymbol{r}}(t_i) \tag{6}$$

is an unbiased estimate of the expected radiance in Equation 3. Moreover, samples $t_1, \ldots, t_k$ come from high-density regions of $p_{\boldsymbol{r}}$ by design, thus for a degenerate density $p_{\boldsymbol{r}}$ even a few samples would provide an estimate with low variance. Each term in Equation 6 contributes equally to the sum.

Importantly, unlike the approximation in Equation 4, the Monte-Carlo estimate depends on scene density $\sigma$ implicitly through sampling algorithm and requires a custom gradient estimate for the parameters of $\sigma$. As an illustration, the full NeRF samples points on a ray from the distribution induced by an auxiliary "coarse" density model. These points are then used as grid knots in approximation 4. However, as their sampling algorithm is non-differentiable and cannot be trained end-to-end, they introduce auxiliary "coarse" radiance field and train "coarse" components separately.

Below, we propose propose a principled end-to-end differentiable algorithm to generate samples from $p_{\boldsymbol{r}}(t)$. We then apply the algorithm to estimate radiance as in Equation 6 and optimize the estimates to reconstruct the density and the radiance field of a scene.

### 3.1 REPARAMETERIZED EXPECTED RADIANCE ESTIMATES

The solution we propose is primarily inspired by the reparameterization trick (12; 31). We first change the variable in Equation 3. For $F_{\boldsymbol{r}}(t) := 1 - \exp\left(-\int_{t_n}^{t} \sigma_{\boldsymbol{r}}(s)\mathrm{d}s\right)$ and $y := F_{\boldsymbol{r}}(t)$ we write

$$C(\boldsymbol{r}) = \int_{t_n}^{t_f} c_{\boldsymbol{r}}(t)p_{\boldsymbol{r}}(t)\mathrm{d}t = \int_{y_n}^{y_f} c_{\boldsymbol{r}}(F_{\boldsymbol{r}}^{-1}(y))\mathrm{d}y. \tag{7}$$

The integral boundaries are $y_n := F_{\boldsymbol{r}}(t_n) = 0$ and $y_f := F_{\boldsymbol{r}}(t)$. Function $F_{\boldsymbol{r}}(t)$ acts as the cumulative distribution function of the variable t with a single exception that, in general, $y_f := F_{\boldsymbol{r}}(t_f) \neq 1$. In volume rendering, $F_{\boldsymbol{r}}(t)$ is called opacity function with $y_f$ being equal to overall pixel opaqueness.

In the right-hand side of Equation 7, the integral boundaries depend on opacity $F_{\boldsymbol{r}}$ and, as consequence, on ray density $\sigma_{\boldsymbol{r}}$. We further simplify the integral by changing the integration boundaries to $[0, 1]$ and substituting $y_n = 0$:

$$\int_{y_n}^{y_f} c_{\boldsymbol{r}}(F_{\boldsymbol{r}}^{-1}(y))\mathrm{d}y = \int_{0}^{1} y_f c_{\boldsymbol{r}}(F_{\boldsymbol{r}}^{-1}(y_f u))\mathrm{d}u. \tag{8}$$

Given the above derivation, we construct **the reparameterized Monte Carlo (R/MC)** estimate for the right-hand side integral in Equation 8 with $k$ i.i.d. $U[0, 1]$ samples $u_1, \ldots, u_k$:

$$\hat{C}_{MC}^{R}(\boldsymbol{r}) := \frac{y_f}{k} \sum_{i=1}^{k} c_{\boldsymbol{r}}(F_{\boldsymbol{r}}^{-1}(y_f u_i)). \tag{9}$$

In the above estimate, random samples $u_1, \ldots, u_k$ do not depend on volume density $\sigma_{\boldsymbol{r}}$ or color $c_{\boldsymbol{r}}$. Essentially, the reparameterized Monte-Carlo estimate generates samples from $p_{\boldsymbol{r}}(t)$ using inverse cumulative distribution function $F_{\boldsymbol{r}}^{-1}(y_f u)$.

We further improve the estimate using stratified sampling. We replace uniform samples $u_1, \ldots, u_k$ with uniform independent samples within regular grid bins $v_i \sim U[\frac{i-1}{k+1}, \frac{i}{k+1}], i = 1, \ldots, k$ and derive **the reparameterized stratified Monte Carlo (R/SMC)** estimate

$$\hat{C}_{SMC}^{R}(\boldsymbol{r}) := \frac{y_f}{k} \sum_{i=1}^{k} c_{\boldsymbol{r}}(F_{\boldsymbol{r}}^{-1}(y_f v_i)). \tag{10}$$

It is easy to show that both 9 and 10 are unbiased estimates of 3. Additionally, the gradient of estimates 9 and 10 is an unbiased estimate of the gradient of the expected color $C(\boldsymbol{r})$. However, in practice we can only query $\sigma_{\boldsymbol{r}}$ at certain ray points and cannot compute $F_{\boldsymbol{r}}$ analytically. Thus, in the following section, we introduce approximations of $F_{\boldsymbol{r}}$ and its inverse.

### 3.2 Opacity Approximations

Expected radiance estimate 9 relies on opacity $F_{\boldsymbol{r}}(t) = 1 - \exp\left(-\int_{t_n}^{t} \sigma_{\boldsymbol{r}}(s)\mathrm{d}s\right)$ and its inverse $F_{\boldsymbol{r}}^{-1}(y)$. We propose to approximate the opacity using a piecewise density field approximation. Figure 2 illustrates the approximations and ray samples obtained through opacity inversion.

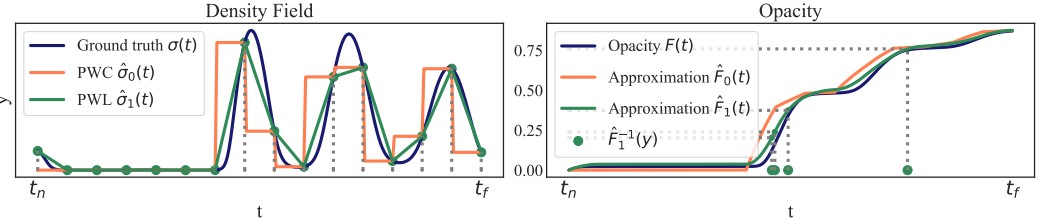

Figure 2: Illustration of opacity inversion. On the left, we approximate density field $\sigma_{\boldsymbol{r}}$ with a piecewise constant (PWC) and a piecewise linear (PWL) approximation. On the right, we approximate opacity $F_{\boldsymbol{r}}(t)$ and compute $F_{\boldsymbol{r}}^{-1}(y_f u)$ for $u \sim U[0, 1]$.

To construct the approximation, we take a grid $t_n = t_0 < t_1 < \cdots < t_m = t_f$ and construct piecewise constant and piecewise linear approximations. In the piecewise linear case, we compute $\sigma_{\boldsymbol{r}}$ in the grid points and interpolate the values between the grid point. In the piecewise constant case, we pick a random point within each bin $t_i \leq \hat{t}_i \leq_t i + 1$ and approximate density with $\sigma_{\boldsymbol{r}}(\hat{t}_i)$ inside the corresponding bin. Importantly, for a non-negative field these approximations are also non-negative.

Then we compute the integral $\int_{t_n}^{t} \sigma_{\boldsymbol{r}}(s)\mathrm{d}s$ used in $F_{\boldsymbol{r}}(t)$ for $t \in [t_i, t_{i+1})$ analytically as a sum of rectangular areas

$$I_0(t) = \sum_{j=1}^{i} \sigma_{\boldsymbol{r}}(\hat{t}_j)(t_j - t_{j-1}) + \sigma_{\boldsymbol{r}}(\hat{t}_i)(t - t_i) \tag{11}$$

for the **the piecewise constant** approximation and as a sum of trapezoidal areas for **the piecewise linear** approximation

$$I_1(t) = \sum_{j=1}^{i} \frac{\sigma_{\boldsymbol{r}}(t_j) + \sigma_{\boldsymbol{r}}(t_{j-1})}{2}(t_j - t_{j-1}) + \frac{(\sigma_{\boldsymbol{r}}(t_i) + \bar{\sigma}_{\boldsymbol{r}}(t))}{2}(t - t_i), \tag{12}$$

where $\bar{\sigma}_{\boldsymbol{r}}(t) = \sigma_{\boldsymbol{r}}(t_i)\frac{t_{i+1}-t}{t_{i+1}-t_i} + \sigma_{\boldsymbol{r}}(t_{i+1})\frac{t-t_i}{t_{i+1}-t_i}$ is the interpolated density at $t$. Given these approximations, we are now able to approximate $F_{\boldsymbol{r}}$ and $y_f$ in Equation 9.

We generate samples on a ray based on inverse opacity $F_{\boldsymbol{r}}^{-1}(y)$ by solving the equation

$$y_f u = F_{\boldsymbol{r}}(t) = 1 - \exp\left(-\int_{t_n}^{t} \sigma_{\boldsymbol{r}}(s)\mathrm{d}s\right) \tag{13}$$

for $t$, where $u \in [0, 1]$ is a random sample. We rewrite the equation as

$$-\log(1 - y_f u) = \int_{t_n}^{t} \sigma_{\boldsymbol{r}}(s)\mathrm{d}s. \qquad (14)$$

and note that integral approximations $I_0(t)$ and $I_1(t)$ are monotonic piecewise linear and piecewise quadratic functions. We obtain the solution of Equation 14 by first finding a bin that contains a solution and then solving a linear or a quadratic equation. Crucially, solution $t$ can be seen as a differentiable function of the density field $\sigma_{\boldsymbol{r}}$ and we can back-propagate the gradients w.r.t. $\sigma_{\boldsymbol{r}}$ through $t$.

In the supplementary materials, we provide explicit formulae for $t$ for both approximations and discuss the solutions crucial for the numerical stability. Additionally, we provide an alternative inversion algorithm in the case when $\int_{t_n}^{t} \sigma_{\boldsymbol{r}}(s)\mathrm{d}s$ can be computed without approximations. In our experiments we report the results only for piecewise linear approximation. In our preliminary experiments, the piecewise constant approximation was faster but delivered worse rendering quality.

## 4 RELATED WORK

There are multiple ways to represent the shape for a scene for novel view synthesis. Earlier learning-based approaches rely on such implicit representations as signed distance fields (27; 34; 35) and occupancy fields (19; 25) to represent non-transparent objects. We concentrate on implicit representations based on density fields pioneered in NeRF (20). Each representation relies on a designated rendering algorithm. In particular, NeRF relies on an emission-absorption optical model developed in (11) with a numerical scheme specified in (18).

**Monte-Carlo estimates for integral approximations.** In this work, we revisit the algorithm introduced to approximate the expected color in (18). Currently, the algorithm is a default solution in multiple of works on neural radiance fields. The authors of (18) approximate density and radiance fields with a piecewise constant functions along a ray and compute 3 as an approximation. Instead, we reparameterize Equation 3 and construct Monte-Carlo estimates for the integral. To compute the estimates in practice we use piecewise approximations only for the density field. The cumulative density function (CDF) used in our estimates involves integrating density field along a ray. In (15), the authors construct field anti-derivatives to accelerate inference. While they use the anti-derivatives to compute 3 on a grid with fewer knots, the anti-derivatives can apply in our framework to construct Monte-Carlo approximations based on the inverse CDF without resorting to piecewise approximations.

In the past decade, integral reparameterizations became a common practice in generative modeling (13; 31) and approximate Bayesian inference (3; 7; 22). Similar to Equation 3, objectives in these areas require optimizing expected values with respect to distribution parameters. We refer readers to (21) for a systematic overview. Notably, in computer graphics, (17) apply reparameterization to estimate gradients of path traced images with respect to scene parameters.

**Algorithms for picking ray points.** Opposed to numerical scheme in Equation 4, our algorithm only requires to evaluate radiance at a sparse set of points sampled from the density field. In (20), the authors use a similar hierarchical scheme to generate ray points using an auxiliary coarse density field. Crucially, unlike our reparameterized importance sampling, the importance sampling algorithm in their work does not allow differentiating with respect to the coarse model parameters. The ad-hoc solution introduced in (20) is to train the coarse model separately using the same rendering objective 5. Subsequent works propose variations of the scheme: Mip-NeRF (2) merges coarse and fine models using a scale-aware neural field, and Mip-NeRF 360 (2) distills the coarse density field from a fine field instead of training an auxiliary coarse radiance field. For non-transparent scenes Unisurf (26) treats the density field as an occupancy field and gradually incorporates root-finding algorithms into volume sampling. Simultaneously, a number of works propose training an auxiliary model to return coarse samples for a given ray. For instance, DoNeRF (24) uses a designated depth oracle network supervised with ground truth depth maps, TermiNeRF (28) foregoes the depth supervision by distilling the sampling network from a pre-trained NeRF model. Finally, the authors of (1) train a proposal network to generate points on a ray end-to-end starting with a pre-trained NeRF. The aforementioned works speed up rendering, but the reliance on auxiliary networks hinders using faster

grid-based architectures and makes the overall scene representation less interpretable. In contrast to the above works, our algorithm learns sampling points on a ray from scratch in an end-to-end fashion, works with an arbitrary density field, and does not requires any auxiliary models.

**NeRF acceleration through architecture and sparsity.** The above algorithms for picking points on a ray generally aim to reduce the number of field evaluations during rendering. An alternative optimization approach is to reduce the time required to evaluate the field. In the past few years, a variety of architectures combining Fourier features (33) and grid-based features was proposed (8; 32; 36; 30). Besides grids, some works exploit space partitions based on Voronoi diagrams (29), trees (10; 37) and even hash tables (23). These architectures generally trade-off inference speed for parameter count. TensorRF (4) stores the grid tensors in a compressed format to achieve both high compression and fast performance. On top of that, skipping the density queries for the empty parts of a scene additionally improves rendering time (14). For the novel view synthesis, the idea allows to speed up rendering during training and inference (9; 6; 16). Notably, our rendering algorithm works with arbitrary density fields and, as a result, is compatible with the improved field architectures and sparse fields.

## 5    EXPERIMENTS

### 5.1    IMPORTANCE SAMPLING FOR A SINGLE RAY

We begin with comparison of importance sampling color estimates in a one-dimensional setting. In this experiment, we assume that we know density in advance and show how the estimate variance depends on number of radiance calls. Compared to importance sampling, the standard approximation from Equation 4 has zero variance but does not allow controlling number of radiance calls.

Our experiment models light propagation on a single ray in three typical situations. The upper row of Figure 3 defines a scalar radiance field (orange) $c_{\boldsymbol{r}}(t)$ and opacity functions (blue) $F_{\boldsymbol{r}}(t)$ for

- "Foggy" density field. It models a semi-transparent volume. Similar fields occur after model initialization during density field training;

- "Glass and wall" density field. Models light passing through nearly transparent volumes such as glass. The light is emitted at three points: the inner and outer surface of the transparent volume and an opaque volume near the end of the ray;

- "Wall" density field. Light is emitted from a single point on a ray. Such fields are most common in applications.

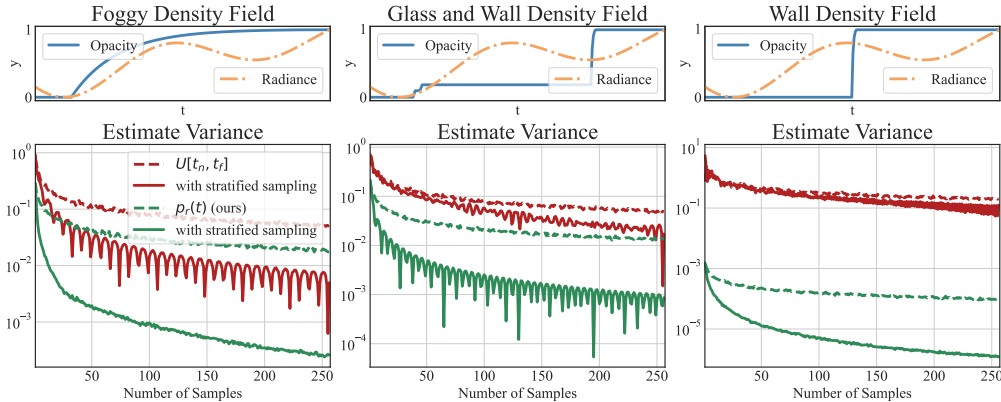

Figure 3: Color estimate variance compared for a varying number of samples. The upper plot illustrates underlying opacity function on a ray; the lower graph depicts variance in logarithmic scale. Compared to a naive importance sampling estimate (dashed red), reparameterized sampling exhibits lower variance (dashed green). Stratified sampling improves variance in both setups (solid lines).

For the three fields we estimated the expected radiance $C(\boldsymbol{r}) = \int_{t_n}^{t_f} c_{\boldsymbol{r}}(t) \mathrm{d}F_{\boldsymbol{r}}(t)$. We considered two baseline methods (both in red in Figure 3): the first was an importance sampling estimate of $C$ obtained with uniform distribution on a ray $U[t_n, t_f]$, and its stratified modification with a uniform grid $t_n = t_0 < \cdots < t_k = t_f$ (note that here we use $k$ to denote the number of samples, not the number of grid points $m$ in piecewise density approximation):

$$\hat{C}_{\mathrm{IW}}(\boldsymbol{r}) = \sum_{i=1}^{k} (t_i - t_{i-1}) c_{\boldsymbol{r}}(\tau_i) \frac{\mathrm{d}F_{\boldsymbol{r}}}{\mathrm{d}t}\bigg|_{t=\tau_i} , \text{ with independent } \tau_i \sim U[t_{i-1}, t_i]. \qquad (15)$$

We compared the baseline against estimate from Equation 9 and its stratified counterpart from Equation 10. All estimates are unbiased. Therefore, we only compared the estimates variances for a varying number of samples $m$.

In all setups, our stratified estimate uniformly outperformed the baselines. For the most challenging "foggy" field, approximately $k = 32$ samples we required to match the baseline performance for $k = 128$. We matched the baseline with only a $k = 4$ samples for other fields. Importance sampling requires only a few points for degenerate distributions. In further experiments, we take $k = 32$ to obtain a precise color estimate even when a model did not converge to a degenerate distribution.

## 5.2 Scene Reconstruction with Reparameterized Volume Sampling

Next, we apply our algorithm to 3D scene reconstruction based on a set of image projections. As a benchmark, we use the synthetic data from NeRF (20). The primary goal of the experiment is to demonstrate computational advantages of our algorithm compared to the basic volume rendering algorithm.

| PSNR (↑) | Chair | Drums | Ficus | Hotdog | Lego | Materials | Mic | Ship | Avg. |
|---|---|---|---|---|---|---|---|---|---|
| NeRF full | 33.00 | 25.01 | 30.13 | 36.18 | 32.54 | 29.62 | 32.91 | 28.65 | 31.01 |
| NeRF w/o h. | 31.32 | 24.55 | 29.25 | 35.24 | 31.42 | 29.22 | 31.74 | 27.73 | 30.06 |
| NeRF (Ours) | 31.35 | 22.42 | 28.42 | 34.36 | 30.70 | 28.72 | 31.18 | 26.89 | 29.26 |
| DVGO | 33.99 | 25.33 | 32.57 | 36.65 | 34.58 | 29.59 | 33.12 | 28.93 | 31.95 |
| DVGO (Ours) | 34.24 | 25.06 | 30.46 | 36.76 | 33.87 | 29.14 | 33.08 | 28.06 | 31.34 |

| SSIM (↑) | Chair | Drums | Ficus | Hotdog | Lego | Materials | Mic | Ship | Avg. |
|---|---|---|---|---|---|---|---|---|---|
| NeRF full | 0.967 | 0.925 | 0.964 | 0.974 | 0.961 | 0.949 | 0.980 | 0.856 | 0.947 |
| NeRF w/o h. | 0.951 | 0.914 | 0.956 | 0.969 | 0.951 | 0.944 | 0.973 | 0.844 | 0.938 |
| NeRF (Ours) | 0.956 | 0.875 | 0.949 | 0.965 | 0.946 | 0.940 | 0.971 | 0.824 | 0.928 |
| DVGO | 0.976 | 0.928 | 0.977 | 0.980 | 0.976 | 0.950 | 0.983 | 0.876 | 0.956 |
| DVGO (Ours) | 0.978 | 0.925 | 0.968 | 0.980 | 0.973 | 0.946 | 0.983 | 0.866 | 0.952 |

| LPIPS (↓) | Chair | Drums | Ficus | Hotdog | Lego | Materials | Mic | Ship | Avg. |
|---|---|---|---|---|---|---|---|---|---|
| NeRF full | 0.046 | 0.091 | 0.044 | 0.121 | 0.050 | 0.063 | 0.028 | 0.206 | 0.081 |
| NeRF w/o h. | 0.065 | 0.177 | 0.056 | 0.130 | 0.072 | 0.080 | 0.039 | 0.249 | 0.109 |
| NeRF (Ours) | 0.065 | 0.178 | 0.066 | 0.078 | 0.083 | 0.077 | 0.040 | 0.225 | 0.102 |
| DVGO | 0.027 | 0.083 | 0.026 | 0.033 | 0.027 | 0.059 | 0.018 | 0.163 | 0.054 |
| DVGO (Ours) | 0.027 | 0.084 | 0.040 | 0.034 | 0.029 | 0.062 | 0.017 | 0.172 | 0.058 |

Table 1: Rendering quality comparison with NeRF and DVGO (32). Metrics are calculated over test views for synthetic scenes (20) with $k = 32$ points in color estimates and $m = 256$ knots along each ray in our NeRF modificatio, for details please see Section 3.2. Our method is slightly worse than NeRF without hierarchical sampling (coarse model) in terms of average PSNR and SSIM, although it is slightly better in terms of average LPIPS. As we have only modified the underlying integration scheme, we expected the model performance to match the non-hierarchical NeRF. For LPIPS calculation we used official implementation (38) and VGG features. Similarly, our modification of DVGO is slightly worse than the original DVGO.

### 5.2.1 NEURAL RADIANCE FIELDS

As our first model, we took the original NeRF's architecture (20) and hyperparameters without any modifications, except for the output activation of field $\sigma$. In particular, we used Softplus activation with $\beta = 10$ instead of ReLU to avoid zero gradients. To form a training batch during training we took a random subset of $64$ training images and sampled $64$ rays per images. To construct a piecewise linear density approximation, we slightly perturbed a uniform ray grid with 256 knots. For the proposed importance-sampling reparameterization we calculated Equation 10 with $k = 32$ samples to estimate color. In the Table 1 below, we report the obtained results. For reference, we also provide the NeRF metrics with and without hierarchical sampling.

With NeRF architecture we expected our algorithm to be comparable to NeRF without hierarchical sampling: these two models use grids of the same density and do not rely on hierarchical sampling. However, despite our expectations, the quantitative results of the baseline were slightly better. The only difference between NeRF without hierarchical sampling and our model is the underlying expected color approximation. We speculate that the variance of our stochastic estimate prevents model from finding fine optimum. For reference we also provide the result for the full NeRF model, however the model is not directly comparable to ours. Even though the full NeRF model also samples points along a ray in a two-stage manner, it re-weights the output points using a second "fine" network, whereas samples in our model are weighted with uniform weights (see Equation 10).

| Model | PSNR ($\uparrow$) | SSIM ($\uparrow$) | LPIPS (VGG)($\downarrow$) | Speed (s)($\downarrow$) | Mem (Gb)($\downarrow$) |
|---|---|---|---|---|---|
| NeRF w/o h. | 31.42 | 0.951 | 0.072 | 26.59 | 6.742 |
| Ours, 1 pt | 25.67 | 0.870 | 0.1768 | 20.18 | 4.189 |
| Ours, 2 pts | 28.29 | 0.909 | 0.1510 | 20.31 | 4.191 |
| Ours, 4 pts | 29.94 | 0.934 | 0.1187 | 20.49 | 4.191 |
| Ours, 8 pts | 30.52 | 0.943 | 0.0930 | 20.82 | 4.191 |
| Ours, 16 pts | 30.68 | 0.946 | 0.0829 | 21.50 | 4.193 |
| Ours, 32 pts | 30.72 | 0.946 | 0.0801 | 22.89 | 4.197 |
| Ours, 64 pts | 30.72 | 0.947 | 0.0794 | 25.74 | 4.209 |

Table 2: Ablation study and comparison in terms of speed and quality with different number of points in improtance-weighted color estimate. We compare inference on views of Lego scene (20). Speed represents the average rendering time of a single $800 \times 800$ frame on NVIDIA v100 GPU. We measured speed and memory usage in pytorch3d's re-implementation of NeRF as our implementation is also written in pytorch. Our algorithm slightly improves rendering time and memory footprint.

Then, we evaluated the proposed method with a varying number of samples at the inference stage ($k$ in Equation 10). We took the Lego scene model from the previous experiment and varied the number of points in our reparametrized color estimation. The quantitative results of this experiments can be found in Table 2 and Figure 4 contain qualitative results. From the rendering quality viewpoint, the three metrics gradually increased with the number of samples and saturated and approximately 16 points. Our algorithm produced sensible renders even for $k = 1$, however noise artifacts only disappeared for $k = 8$.

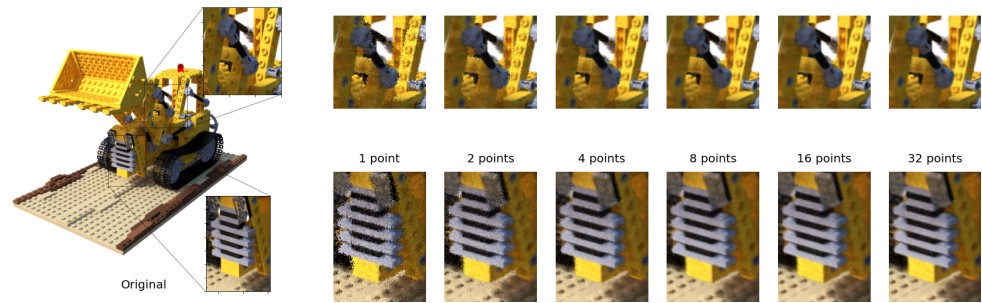

Figure 4: NeRF rendering results with a different number of samples in the proposed stratified estimate with re-sampling. From left to right and from top to down: 1, 2, 4, 8, 32 points estimates and ground truth for reference.

### 5.2.2 DIRECT VOXEL GRID OPTIMIZATION

We also tested our rendering algorithm on a recent voxel-based radiance field model DVGO (32). The model takes only a few minutes to train thanks to lightning fast architecture, progressive scaling, and custom CUDA kernels. We took the official implementation and only replaced the rendering algorithm based on Equation 4. To achieve the rendering performance competitive with their CUDA kernels, we optimized the performance using just-in-time compilation module in Pytorch.

We evaluated the two training stages of DVGO with a varying number of radiance samples. On the first "coarse" stage, the model fits a low-resolution 3D density grid and a view-independent 3D radiance grid. On the second "fine" stage, the model fits a density grid with gradually improving resolution and view-dependent radiance field combining a grid and an MLP. Crucially, the second stage relies on a coarse grid from the first stage to skip empty regions in space and optimize the performance. Table 3 presents the comparison results on a synthetic Lego scene. Results for other synthetic scenes for $k = 32$ samples are in Table 1.

| | Radiance calls ↓ | | Mem (Gb) ↓ | | Training time (s) ↓ | | PSNR |
| | Coarse | Fine | Coarse | Fine | Coarse | Fine | |
|---|---|---|---|---|---|---|---|
| Ours | 1 | 1 | 4.7 | 5.0 | 39 | 167 | 29.84 |
| Ours | 8 | 8 | 4.7 | 5.1 | 39 | 205 | 33.29 |
| Ours | 32 | 32 | 4.7 | 5.7 | 40 | 362 | 33.95 |
| Ours | 64 | 64 | 4.7 | 7.3 | 42 | 581 | 34.15 |
| DVGO | 130.7 | 13.4 | 4.7 | 5.5 | 54 | 238 | 34.60 |

Table 3: Comparison between the default rendering algorithm and the proposed reparameterized estimates with a varying number of samples for DVGO model. Our algorithm improves training speed on the coarse stage by up to $30\%$ and can improve training speed on the fine stage at cost of lower model quality.

On the coarse stage, we observed at least $20\% - 30\%$ improvement in training time. The improvement can be attributed to fewer radiance samples. Using the auxiliary mask for empty space regions, DVGO significantly reduces the number of radiance calls on the second stage. As a result, our rendering algorithm improved training time only when the number of samples is lower than the average number of radiance calls in DVGO (13.4 in this case). At the same time, as the PSNR column indicates, the rendering quality deteriorated with fewer radiance samples. Notably, in our experiments, the model trained with 64 samples achieved PSNR 34.33 even with 16 during evaluation stage. We concluded that lower PSNRs are caused by the estimate variance during training.

## 6 CONCLUSION

In this work, we proposed an alternative rendering algorithm for novel view synthesis models based on radiance fields. The core of our contribution is end-to-end differentiable ray point sampling algorithm. For two pre-existing architectures, we show that the algorithm can achieve competitive rendering quality while reducing training and rendering time, and required GPU memory. Besides that, we believe that such an algorithm opens up new possibilities in efficient rendering and architecture design that are yet to be explored.

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

# A    APPENDIX

Below we discuss caveats and implementation details of our sampling algorithm.

## A.1    INVERSE FUNCTIONS FOR DENSITY INTEGRALS

In this section, we derive explicit formulae for the density integral inverse used in inverse opacity.

### A.1.1    PIECEWISE CONSTANT APPROXIMATION INVERSE

We start with a formula for the integral

$$I_0(t) = \sum_{j=1}^{i} \sigma_{\boldsymbol{r}}(\hat{t}_j)(t_j - t_{j-1}) + \sigma_{\boldsymbol{r}}(\hat{t}_i)(t - t_i) \tag{16}$$

and solve for $t$ equation

$$y = I_0(t). \tag{17}$$

The equation above is a linear equation with solution

$$t = t_i + \frac{y - \sum_{j=1}^{i} \sigma_{\boldsymbol{r}}(\hat{t}_j)(t_j - t_{j-1})}{\sigma_{\boldsymbol{r}}(\hat{t}_i)}. \tag{18}$$

In our implementation we add small $\epsilon$ to the denominator to improve stability when $\sigma_{\boldsymbol{r}}(\hat{t}_i) \approx 0$.

### A.1.2    PIECEWISE LINEAR APPROXIMATION INVERSE

The piecewise linear density approximation yield a piecewise quadratic function

$$I_1(t) = \sum_{j=1}^{i} \frac{\sigma_{\boldsymbol{r}}(t_j) + \sigma_{\boldsymbol{r}}(t_{j-1})}{2}(t_j - t_{j-1}) + \frac{(\sigma_{\boldsymbol{r}}(t_i) + \bar{\sigma}_{\boldsymbol{r}}(t))}{2}(t - t_i). \tag{19}$$

Again, we solve

$$y = I_1(t) \tag{20}$$

for $t$. We change the variable to $\Delta t := t - t_i$ and note that terms $a$ and $c$ in quadratic equation

$$0 = a\Delta t^2 + b\Delta t + c \tag{21}$$

will be

$$a = \frac{\sigma_{\boldsymbol{r}}(t_{i+1}) - \sigma_{\boldsymbol{r}}(t_i)}{2} \tag{22}$$

$$c = \left( \sum_{j=1}^{i} \frac{\sigma_{\boldsymbol{r}}(t_j) + \sigma_{\boldsymbol{r}}(t_{j-1})}{2}(t_j - t_{j-1}) - y \right) \times (t_{i+1} - t_i) \tag{23}$$

and with a few algebraic manipulations we find the linear term

$$b = \sigma_{\boldsymbol{r}}(t_i) \times (t_{i+1} - t_i). \tag{24}$$

Since our integral monotonically increases, we can deduce that the root $\Delta t$ must be

$$\Delta t = \frac{-b + \sqrt{b^2 - 4ac}}{2a}. \tag{25}$$

However, this root is computationally unstable when $a \approx 0$. The standard trick is to rewrite the root as

$$\Delta t = \frac{2c}{b + \sqrt{b^2 - 4ac}}. \tag{26}$$

For computational stability, we add small $\epsilon$ to the square root and denominators use replace the root with $\Delta t = \sqrt{\frac{a}{c}}$ when $b \approx 0$ and $\Delta t = -\frac{c}{b}$ when $a \approx 0$. See the supplementary notebook for details.

## A.2 Numerical Stability in Inverse Opacity

Inverse opacity input $y$ is a combination of a uniform sample $u$ and ray opacity $y_f = 1 - \exp\left(-\int_{t_n}^{t_f} \sigma_{\boldsymbol{r}}(s)\mathrm{d}s\right)$:

$$y = -\log(1 - y_f u). \tag{27}$$

The expression above is a combination of a logarithm and exponent. We rewrite it to replace with more reliable logsumexp operator:

$$y = -\log\left(\exp(\log(1 - u)) + \exp\left(\log u - \int_{t_n}^{t_f} \sigma_{\boldsymbol{r}}(s)\mathrm{d}s\right)\right). \tag{28}$$

In practice, for opaque rays $\int_{t_n}^{t_f} \sigma_{\boldsymbol{r}}(s)\mathrm{d}s \approx 0$ implementation of logsumexp becomes computationally unstable. In this case, we replace $y$ with $u$ as they are almost identical.

## A.3 Implicit Inverse Opacity Gradients

To compute the estimates in Equation 9, we need to compute the inverse opacity $F_{\boldsymbol{r}}^{-1}(y)$ along with its gradient. In the main paper, we invert opacity explicitly with a differentiable algorithm. Alternatively, we could invert $F_{\boldsymbol{r}}(t) = 1 - \exp\left(-\int_{t_n}^{t} \sigma_{\boldsymbol{r}}(s)\mathrm{d}s\right)$ with binary search.

Opacity $F_{\boldsymbol{r}}(t)$ is a monotonic function and for $y \in (y_n, y_f) = (F_{\boldsymbol{r}}(t_n), F_{\boldsymbol{r}}(t_f))$ the inverse lies in $(t_n, t_f)$. To compute $F_{\boldsymbol{r}}^{-1}(y)$, we start with boundaries $t_l = t_n$ and $t_r = t_f$ and gradually decrease the gap between the boundaries based on the comparison of $F_{\boldsymbol{r}}(\frac{t_l + t_r}{2})$ with $y$. Importantly, such procedure is easy to parallelize across multiple inputs and multiple rays.

However, we cannot back-propagate through the binary search iterations and need a workaround to compute the gradient $\frac{\partial t}{\partial \theta}$ of $t(\theta) = F_{\boldsymbol{r}}^{-1}(y, \theta)$. To do this, we follow (5) and compute differentials of the right and the left hand side of equation $y(\theta) = F_{\boldsymbol{r}}(t, \theta)$

$$\frac{\partial y}{\partial \theta}\mathrm{d}\theta = \frac{\partial F_{\boldsymbol{r}}}{\partial t}\frac{\partial t}{\partial \theta}\mathrm{d}\theta + \frac{\partial F_{\boldsymbol{r}}}{\partial \theta}\mathrm{d}\theta. \tag{29}$$

By the definition of $F_{\boldsymbol{r}}(t, \theta)$ we have

$$\frac{\partial F_{\boldsymbol{r}}}{\partial t} = (1 - F_{\boldsymbol{r}}(t, \theta))\sigma_{\boldsymbol{r}}(t, \theta), \tag{30}$$

$$\frac{\partial F_{\boldsymbol{r}}}{\partial \theta} = (1 - F_{\boldsymbol{r}}(t, \theta))\frac{\partial}{\partial \theta}\left(\int_{t_n}^{t} \sigma_r(s, \theta)\mathrm{d}s\right). \tag{31}$$

We solve Equation 29 for $\frac{\partial t}{\partial \theta}$ and substitute the partial derivatives using Equations 30 and 31 to obtain the final expression for the gradient

$$\frac{\partial t}{\partial \theta} = \frac{\frac{\partial y}{\partial \theta} - (1 - F_{\boldsymbol{r}}(t, \theta))\frac{\partial}{\partial \theta}\int_{t_n}^{t} \sigma_{\boldsymbol{r}}(s, \theta)\mathrm{d}s}{(1 - F_{\boldsymbol{r}}(t, \theta))\sigma_{\boldsymbol{r}}(t, \theta)}. \tag{32}$$

In our implementation, we use automatic differentiation to compute $\partial y/\partial \theta$ and $\frac{\partial}{\partial \theta}\int_{t_n}^{t} \sigma(s)\mathrm{d}s$ to combine the results as in Equation 32.

