# OpenReview forum: "Differentiable Rendering with Reparameterized Volume Sampling"
_ICLR.cc/2023/Conference — Submitted to ICLR 2023_

### Official Review · Reviewer_D4iJ · 2022-10-18

**Confidence:** 4
**Correctness:** 2
**Technical Novelty And Significance:** 1
**Empirical Novelty And Significance:** 1
**Recommendation:** 3

**Clarity, Quality, Novelty And Reproducibility:**

I found no significant problems in these respects. The paper itself is well written and easy to follow. The proposed idea is clearly presented. I think the authors made a reasonable effort to include sufficient information for reproduction.

The author' idea, the use of accurate importance sampling for a neural radiance field, is somewhat new in the sense that  I do not know the same work that has been done/published before. However, as I mentioned in "Strength and Weaknesses", I do not think that this idea (in the current form) is reasonable and useful for advancing the technology in this field.

**Details Of Ethics Concerns:**

No specific concerns.

**Strength And Weaknesses:**

Strengths

- The idea of importance sampling is explained well in a logical manner. As far as I read, the theory and the related derivations are correct. Moreover, this idea was successfully implemented as a trainable module that can be incorporated into the existing neural rendering methods.

Weaknesses

- The authors failed to establish the effectiveness (computational advantages over the previous methods) of their proposal.  As shown in Tables 2 and 3, the authors' method achieved only a small amount (up to 25 percent) of reduction in computation time  compared to the baselines (NeRF and DVGO) at the cost of quality decline. The authors should notice the fact that the number of sampling points can be changed also for the original methods (NeRF and DVGO) so as to find a desirable trade-off between the computational cost and rendering quality. However, the authors used a fixed number of samples for the baselines,  and thus, I cannot see the advantage of the authors' proposal over the baseline methods. Moreover, the authors did not compare their proposed method against other speed-up methods for neural radiance fields. For example, DONeRF achieved a significant improvement in quality-speed trade-off: 15-78 times fewer computation and comparable rendering quality than/to the baseline NeRF. KiloNeRF achieved more than 2000 times speed up while maintaining the rendering quality. Compared to these works, the contribution of the present submission seems quite limited.

- Insufficient consideration/evaluation for the method's design. The proposed method was designed to reduce the number of "radiance queries", but it still needs many "density queries" to obtain accurate opacity along the light ray. Moreover, the re-parameterization for the importance sampling introduced additional computational cost. Therefore, it seems that the authors' method cannot save the computation so much as the other speed-up methods. The authors should evaluate computational cost (e.g. in FLOP per pixel) involved in each of these steps (density queries, re-parameterization, and radiance queries) to see the problem clearer. Moreover, it seems that the authors sticked to the idea of using a single radiance field, rejecting the use of auxiliary/multiple networks. However, the use of auxiliary/multiple networks has practical advantages in terms of the speed-quality trade-off.

**Summary Of The Paper:**

The authors proposed a rendering method for a neural radiance field. When marching on each light ray, the authors chose the sampling points using the idea of importance sampling; the samples are drawn in accordance with the contributions to the final rendered color. The method was designed so as to reduce the number of the radiance queries, which (they expected) would bring the computational advantages over the previous methods. They incorporated this sampling strategy into two baseline methods, NeRF and DIGO, and compared the performance with the original methods.

**Summary Of The Review:**

As mentioned in "Strength and Weaknesses", I appreciate the logical explanation for the importance sampling for a neural radiance field. However, I think the authors' contribution is insufficient in the current form; they have not established the effectiveness of their proposal: computational advantages over the previous methods. Therefore, I recommend rejection for this submission.

---

> ### Author Response · Authors · 2022-11-18
> **Reply to reviewer D4iJ**
>
> **The authors failed to establish the effectiveness (computational advantages over the previous methods) of their proposal.
> ... For example, DONeRF achieved a significant improvement in quality-speed trade-o: 15-78 times fewer computation and comparable rendering quality than/to the baseline NeRF. KiloNeRF achieved more than 2000 times speed up while maintaining the rendering quality.**
>
> Our primary goal was to demonstrate the benefits of the proposed rendering algorithm. The rendering speed improvement is a side-effect of the algorithm.
>
> There already exists a variety of works that speed up the vanilla NeRF and, as the reviewer pointed out, some of these works already speed up the vanilla algorithm by orders of magnitude. The community pointed out various improvement directions: faster architectures, skipping space regions, early termination, and lower-level implementations. However, all works use the same sum for rendering and in this respect of work is novel.
>
> In our experiments, we took two typical architectures. Nerf is the simplest architecture that can be considered too slow by the current standards. DVGO is a recent grid-based approach that combines the tricks used to speed up NeRFs. For example, DVGO trains approximately 300 times faster than NeRF. Without tweaking any hyperparameters or modifying architecture, we improve rendering speed by up to 25% both in vanilla NeRF, a slow architecture by current standards, and in carefully optimized DVGO.
>
> In our experiments, we controlled for the rendering algorithm and intentionally avoided direct comparison with some of the recent works to eliminate other factors influencing the performance. For instance, DONeRF uses an additional oracle network, and KiloNeRF relies on distillation for training. Still, we agree that we should have compared model training time at the same PSNR level. We will add a summary of key performance metrics of recent approaches for reference.
>
> **The proposed method was designed to reduce the number of "radiance queries", but it still needs many "density queries" to obtain accurate opacity along the light ray. Moreover, the re-parameterization for the importance sampling introduced additional computational cost. Therefore, it seems that the authors' method cannot save the computation so much as the other speed-up methods. The authors should evaluate computational cost [...] involved in each of these steps (density queries, re-parameterization, and radiance queries) to see the problem clearer.**
>
> Thank you for the suggestion. We put a significant effort into optimizing our sampling algorithm but did not explicitly estimate runtime. There is a tiny overhead, but it is diminishing compared to the field evaluations. As a result, we obtained marginal improvements in seconds per iteration both in multilayer MLPs and semi-explicit fields in DVGO.
>
> For a single pixel and $n$ points on a ray, the standard rendering algorithm runs several component-wise operations and computes a cumulative sum. This yields asymptotic complexity O(m) for time and memory. Our algorithm also uses several component-wise transformations, computes a cumulative sum, and additionally finds a bin on a ray containing each sample to obtain $m$ samples. This yields asymptotic complexity O(m + k log(m)) for time and O(m + k) for memory.
>
> Due to the point-wise operations, the two rendering algorithms are memory-bound, and FLOPs do not reflect their actual performance. Therefore, we compare iteration times for a batch of $b = 2048$ rays and $m = 256$ grid points below. Due to parallelization, the compute time in our algorithm does not significantly change with the number of samples $k$. We take $k=32$ and report average iteration times (in seconds per batch) for training (forward + backward) and inference (forward only).
>
> |           | Baseline Rendering |  Our Rendering   | NeRF Backbone | DVGO Density Grid | DVGO Hydrid Radiance |
> |-----------|--------------------|------------------|---------------|-------------------|----------------------|
> |Train      |    0.00106         |   0.00251        |  0.04364      |   0.00033         |    0.02868           |
> |Inference  |    0.00033         |   0.00082        |  0.02407      |   0.00002         |    0.02608           |
>
> The baseline rendering algorithm is several times faster than our rendering algorithm, yet both are orders of magnitude faster than NeRF backbone network and the hybrid radiance model in DVGO.
>
> In our NeRF experiments, we computed the backbone for $m + k$ points and saved computation by evaluating the radiance (a 3-layer MLP on top of the backbone) only for $k$ samples.
>
> In DVGO, radiance evaluation takes significantly more time compared to density. The original DVGO algorithm prunes ray points with low weight, which leads to an average of 13 radiance calls per ray. Our rendering is faster when we use fewer points. Without pruning, the coarse training stage of DVGO takes 809 seconds instead of 238.

---

### Official Review · Reviewer_XnSd · 2022-10-24

**Confidence:** 5
**Correctness:** 3
**Technical Novelty And Significance:** 2
**Empirical Novelty And Significance:** 2
**Recommendation:** 3

**Clarity, Quality, Novelty And Reproducibility:**

* **Clarity**: Most contents are clear and easy to follow.
* **Quality**: The proposed method has not yet been evaluated and studied on substantial benchmarks and settings (refer to weaknesses).
* **Novelty**: The proposed method is somewhat new but has some points in common with previous ones, e.g. density estimation.
* **Reproducibility**: Most implementation details are provided and a script is provided. It is likely to reproduce
the method easily.

**Strength And Weaknesses:**

**Strength**
The topic of designing better sampling methods for volume rendering is interesting and valuable. This paper proposes a reparameterization method to enable end-to-end differentiable optimization, which is reasonable and has been evaluated in NeRF-provided benchmarks.

**Weaknesses**
**The biggest concerns come from the experimental evaluation and comparisons.**
* The proposed method shows a worse rendering quality while the speedup over the original NeRF and DVGO is not so significant.
* Related works, e.g. NeRF-ID, DO-NeRF, AutoInt, TermiNeRF, should also be compared in experiments with regard to both rendering quality and speed.
* Some highly related works are missing, including but not limited to [1, 2]. Please have a careful check.
* Experimental evaluation is not so comprehensive. Real scenes, e.g. LLFF ones, need to be included.

**Minor**:
* The citation style does not follow the ICLR-required author-year ones.

[1] Fang J, Xie L, Wang X, et al. Neusample: Neural sample field for efficient view synthesis[J]. arXiv preprint arXiv:2111.15552, 2021.
[2] Kurz A, Neff T, Lv Z, et al. AdaNeRF: Adaptive Sampling for Real-time Rendering of Neural Radiance Fields[J]. arXiv preprint arXiv:2207.10312, 2022.


**Summary Of The Paper:**

Volume rendering is a critically important algorithm in NeRF (neural radiance field) methods, which connects 3D real points with 2D image pixels. This paper aims at exploring an alternative rendering algorithm mainly focusing on the importance sampling for integration. A reparameterized Monte Carlo estimation method is proposed to enable end-to-end optimization, where fewer samples are needed for integration to save cost.

**Summary Of The Review:**

The main concerns lie in experimental evaluation and comparisons. For its current state, no evident advantages over previous works are shown.

---

> ### Author Response · Authors · 2022-11-18
> **Reply to reviewer XnSd**
>
> **The proposed method shows a worse rendering quality while the speedup over the original NeRF and DVGO is not so significant.**
>
> Indeed, we observed that the proposed estimate leads to worse PSNR on average in our experiments. However, for several scenes, our estimate leads to better PSNR.
>
> Regarding the performance, there already exists a line of work directly addressing NeRF efficiency. Some of these works improve critical performance aspects (e.g., training time, rendering speed) by several orders of magnitude, so our speedup may not be as impressive. But we want to emphasize that the presented rendering algorithm finds room for improvement in the vanilla NeRF and DVGO, a model with carefully optimized performance (training in minutes rather than days).
>
> **Related works, e.g. NeRF-ID, DO-NeRF, AutoInt, TermiNeRF, should also be compared in experiments with regard to both rendering quality and speed.**
>
> We will add a careful comparison in a future revision. Please note that all aforementioned works directly address rendering time. As a result, these works either modify the architecture (as in AutoInt) or require an auxiliary network (NeRF-ID, Do-NeRF, TermiNeRF) with a dedicated training stage. This often leads to inference time improvements at cost of longer training time.
>
> On the opposite side, our work only modifies the expected color estimate and requires no additional training stages or architecture modifications. The low number of samples allows us to improve both training and rendering time. For a fair comparison, we will need to collect data on the total training time of the aforementioned approaches.
>
> **Some highly related works are missing, including but not limited to [1, 2]. Please have a careful check.**
>
> Thank you for pointing out these works. We will discuss them in a future revision.
>
> **Experimental evaluation is not so comprehensive. Real scenes, e.g. LLFF ones, need to be included.**
>
> We will add an evaluation on real scenes in a future revision. However, we do not expect any qualitative differences with the results on synthetic scenes as we only replace the expected color approximation.
>
> **Novelty: The proposed method is somewhat new but has some points in common with previous ones, e.g. density estimation.**
>
> Could you please elaborate on what you mean by density estimation in this case?

---

### Official Review · Reviewer_yzB7 · 2022-10-24

**Confidence:** 4
**Clarity, Quality, Novelty And Reproducibility:** The originality is good. Clarity is f…
**Correctness:** 3
**Technical Novelty And Significance:** 2
**Empirical Novelty And Significance:** 2
**Recommendation:** 5

**Strength And Weaknesses:**

Strength:
1) The mathematical analysis looks great.

Weakness:
1)	The proposed method can lead to improvements on efficiency but degrades on quality. But the experiments presented in the paper are hard to support the claimed contribution. It would be fair if make experiments of evaluating two methods on the time used of reaching the same PSNR. Or vice versa, evaluating the produced PNSR with the same training time.
2)	The improved efficiency is very slight compared with baseline approach. Normally, a smaller number of evaluations to MLPs should lead to great efficiency improvements. But in this paper, the efficiency improvement is little. The author should analyze the reason deeply, is this caused by the extra time used on solving Eq. 14 and 10?
3)	The reasons of piece wise constant has worse quality than piece wise linear are not analyzed.
4)	The reasons of different performance on three experiment setups (Fig. 3) are not analyzed.
5)	“Notably, in our experiments, the model trained with 64 samples achieved PSNR 34.33 even with 16 during evaluation stage.” I cannot find the 34.33 in paper.


**Summary Of The Paper:**

This paper aims at avoiding computing radiance in less contributive parts by reparameterizing the sampling algorithm. This can help decrease the number of evaluations to MLPs. This sounds reasonable. But the experiments do not meet the authors’ expectations. First, decreasing the MLP evaluations should indeed improve efficiency a lot. But the results show only minor improvements on NeRF baseline. This reminds us of if the proposed approach itself slows down the efficiency. Second, DVGO is an explicit volume-based approach which is extremely efficient. I have to say the author have chosen a wrong baseline. Because the efficiency of explicit volume-based approach is not severely slow down by the great number of sampling points. Third, a big number of sampling points is vital to rendering quality. So, the rendering quality of this paper degrades. Fourth, the experiments cannot support the claim. Please refer to the weakness for more details.

**Summary Of The Review:**

The unimpressive experiment results mostly lead to a negative opinion to this paper.
The claimed contribution on efficiency cannot be supported by the experiment. Some constraints (time used on reaching the same psnr, or evaluating the produced psnr using the same training time) have to be set for a fair comparison. And the rendering quality is also not good.

---

> ### Author Response · Authors · 2022-11-18
> **Reply to reviewer yzB7**
>
> **The proposed method can lead to improvements on efficiency but degrades on quality. ... It would be fair if make experiments of evaluating two methods on the time used of reaching the same PSNR. Or vice versa, evaluating the produced PNSR with the same training time.**
>
> We agree that one of the two parameters should be fixed and will improve the protocol in a future revision. Qualitatively, with fewer samples, our algorithm converges to a worse local optimum with more floater artifacts. Increasing the number of iterations of our algorithm to level training time does not increase PSNR.
>
> Even though we struggle to find an edge on a full-scale model, during the coarse training stage of DVGO with our rendering algorithm, we find equally good coarse scene approximations. For instance, on the lego scene with $k=8$ samples our method finds a coarse approximation with 23.89 PSNR in 36 seconds. Then, based on that grid, the default fine stage of DVGO achieves 34.52 PSNR. The default coarse stage of DVGO achieves 23.97 PSNR in 53 seconds. Then, based on that grid DVGO achieves 34.50 PSNR on the fine stage. In other words, the coarse grid PSNR is slightly off, but it does not affect the resulting model performance. Another possible use-case of our algorithm is interactive model visualization, where a slight drop in rendering quality is acceptable in return for faster rendering.
>
> **The improved efficiency is very slight compared with baseline approach. Normally, a smaller number of evaluations to MLPs should lead to great efficiency improvements. But in this paper, the efficiency improvement is little. The author should analyze the reason deeply, is this caused by the extra time used on solving Eq. 14 and 10?**
>
> Time required to solve Eq. 14 and 10 compromises only a tiny portion of rendering time. On average, our importance sampling is 2-3 times slower than the default rendering algorithm, but still an order of magnitude faster than the other components  (our response to reviewer D4iJ has additional details). There is another explanation for the observed performance.
>
> In our experiments, we did not change the architecture of NeRF and the original architecture was designed to compute density and radiance simultaneously. In particular, NeRF consists of a joint backbone with a head for density and a head for radiance on top of the backbone. With the standard rendering algorithm and $m$ grid knots, NeRF runs the backbone and both heads $m$ times. With the same architecture and $k$ samples, our algorithm evaluates the backbone $m + k$ times, evaluates the density head $m$ times, and the radiance head $k$ times. The marginal improvements come from fewer radiance head evaluations.
>
> As a proof of concept, we ran an experiment with NeRF architecture adapted to our algorithm. Specifically, we split the joint 8-layer backbone into two parts of equal size: one part is used to compute density features, and another part is used to compute radiance features. With the smaller network required to compute density the speedup factor increases to 2.3. However, without carefully tuning the network sizes we obtained a suboptimal ~30 PSNR for the baseline and our algorithm.
>
> | Rendering | Density Calls | Radiance Calls | Sec. / It. |
> |-----------|---------------|----------------|------------|
> |  Baseline | 256           | 256            | 27.63      |
> |    Ours   | 256           | 1              | 11.34      |
> |    Ours   | 256           | 2              | 11.39      |
> |    Ours   | 256           | 4              | 11.48      |
> |    Ours   | 256           | 8              | 11.67      |
> |    Ours   | 256           | 16             | 12.10      |
> |    Ours   | 256           | 32             | 13.06      |
> |    Ours   | 256           | 64             | 14.98      |

---

> > ### Author Response · Authors · 2022-11-18
> > **...and the second part**
> >
> > **Third, a big number of sampling points is vital to rendering quality. ... I cannot find the 34.33 in paper.**
> >
> > We did not put the observation in any separate table and only mentioned it in the main text. In our experiments, we observed that with fewer samples the model tends to find worse local optimum. Therefore, a big number of sampling points is vital to optimisation. But after finding near-optimal fields, the algorithm produces high-quality renders even with several samples. We took the model trained with 64 samples and even observed a slight increase in PSNR (from 34.15 to 34.44) with 16 samples. To double check,
> >
> > **3) The reasons of piece wise constant has worse quality than piece wise linear are not analyzed**
> >
> > Intuitively, a continuous piecewise linear approximation must be a better approximation for the continuous density field compared to a discontinuous piecewise constant approximation. We will add an empirical comparison in a future revision.
> >
> > **4) The reasons of different performance on three experiment setups (Fig. 3) are not analyzed.**
> >
> > In all experiments, sampling with inverse density function yields lower variance than uniform sampling from the domain. For degenerate distributions, the difference gets more drastic. The difference occurs because most of the time uniform sampling returns zero-weight samples and occasionally finds points with very high density. At the same time, importance sampling always returns points from high-density regions with equal weights. Stratified sampling is another common trick used to reduce the variance of a Monte Carlo estimate. Importance sampling and stratified sampling generally reduce estimate variance. However, in some corner cases, they may not help.
> >
> > These performance differences are typical for Monte-Carlo estimates. In Figure 3, we illustrated and quantified the differences for several representative examples and chose an appropriate number of samples for further experiments. What kind of analysis would be appropriate beside the qualitative reasoning from the above?

---

### Official Review · Reviewer_bm7A · 2022-10-25

**Confidence:** 4
**Correctness:** 4
**Technical Novelty And Significance:** 3
**Empirical Novelty And Significance:** 1
**Recommendation:** 5

**Clarity, Quality, Novelty And Reproducibility:**

The paper is well written. There are some typos ("improtance" in the Table 2 section) and some grammatical errors, but these don't hurt understanding of the paper. The contribution is not very novel, given that NeRF does point out the need for some form of importance sampling, and implements an effective ad-hoc importance sampling-style scheme.

**Strength And Weaknesses:**

The paper is well written and well motivated. The description of the technical contribution is clear, and Figures 1-3 are good visualizations of the technique.

The outlined idea holds some promise, given that NeRFs coarse-then-fine scheme, despite its effectiveness, is not thoroughly analyzed in the original paper.

A drawback of the proposed technique is the need to finely sample the density field to create the approximation for inverse transform sampling. This seems to preclude major performance benefits that would otherwise be associated with importance sampling.

The experiments reported in Section 5.2 suggest that the proposed importance sampling scheme just doesn't work very well in its current implementation. The reconstruction quality is consistently lower than NeRF's, even if the coarse network of NeRF is disabled. (Could the authors clarify how many radiance evaluations the NeRF models in Table 1 and 2 perform?)

The comparison to DVGO also fails to show significant performance or quality improvements.


**Summary Of The Paper:**

The paper proposes an importance sampling scheme for the radiance evaluation of NeRF-style volumetric rendering. Importance sampling replaces the coarse network in NeRF with direct sampling of the high resolution density field. This promises a simpler algorithm without the need for a proxy loss on the coarse network, and improved quality/performance, as radiance samples are concentrated in regions of high opacity, while transparent regions are skipped.

The paper describes the relevant background on volumetric rendering and outlines how the piecewise constant ray integral can be recast, first as a monte-carlo estimate in ray space, and then as as an integral over the normalized opacity along the ray.

The paper describes in detail how the approximate opacity is computed with a piecewise approximation, and how samples from the distribution are drawn via inverse transform sampling. A complication of the paper is that in order to construct an approximation of the cumulative density function, the technique does require a relatively fine (256) sampling of the density field, so any performance improvement of importance sampling would be gained solely by saving radiance evaluations.

There is a insightful synthetic experiment that compares the proposed integration scheme to uniform sampling in ray space (with ablations for stratified sampling).

The paper then presents comparisons to NeRF and a voxel-based technique (DVGO). It appears from these experiments that both baselines (full NeRF with coarse model, and an unguided NeRF without hierarchical sampling) perform better than the proposed technique.



**Summary Of The Review:**

I'm leaning against accepting this paper. The empirical evaluation is unfortunately not very impressive. The technical content is solid, but only addresses a narrow aspect of the overall algorithm, and does not include enough new ideas to put the paper over the bar in light of the mixed experimental results.

---

> ### Author Response · Authors · 2022-11-18
> **Reply to reviewer bm7A**
>
> **A drawback of the proposed technique is the need to finely sample the density field to create the approximation for inverse transform sampling. This seems to preclude major performance benefits that would otherwise be associated with importance sampling.**
>
> Yes, the described method requires constructing an accurate density approximation. For the models with expensive density evaluation such as vanilla NeRF, this precludes any significant performance benefits. On the other hand, in some models density evaluation is less expensive. For instance, querying the explicit density field in DVGO is two orders of magnitude faster than querying the vanilla NeRF. At the same time, the hybrid radiance field in DVGO is not as fast, and reducing the number of radiance evaluations significantly affects the performance.
>
> In Table 3, we compare our method against the default DVGO model that skips radiance evaluations based on several ad-hoc rules. Without these skipping rules, the same number of iterations during the fine stage takes 809 seconds instead of 238. Our sampling scheme can be seen as an alternative to these ad-hoc rules that reduces iteration cost even further with fewer radiance evaluations.
>
> **Could the authors clarify how many radiance evaluations the NeRF models in Table 1 and 2 perform?**
>
> Our models reported in Table 1 use $k=32$ samples during training and evaluation, resulting in 32 radiance calls per ray. NeRF with and without hierarchical sampling computes radiance and density at $m=256$ points at each ray. In Table 2, we take a model trained with $k=32$ samples and use a varying $k$ for inference.
>
> Counting radiance evaluations in DVGO is less trivial. On the one hand, DVGO gradually increases resolution during training. As a result, the number of summands in Eq. 4 and radiance calls can increase over time. On the other hand, DVGO skips summands with low weights to optimize the performance. Skipping free space results in at least 2x speed up in DVGO (table I4 in DVGO studies how skipping affects the performance, we also discuss the issue in our response to reviewer D4iJ). The number of radiance calls depends on a scene, model hyperparameters, and the current ray. For example, on average, DVGO evaluates radiance 13 times per ray on the Lego scene.

---

### Author Response · Authors · 2022-11-18
**To all**

We thank the reviewers for their time and insightful comments. Below, we individually address some of the points in each review. We did not revise the submission hoping to address some reviewer points more carefully in the future.

---

### Decision · Program_Chairs · 2023-01-20

**Decision:**

Reject

**Justification For Why Not Higher Score:**

see meta-review above

**Justification For Why Not Lower Score:**

see meta-review above

**Metareview: Summary, Strengths And Weaknesses:**

This work focuses on a novel proposed method for differentiable rendering, aiming for a computational speed-up.

According to my evaluation and the remarks of the reviewers, the main strengths of this paper are its clarity and motivation, as well as the novelty of the method. The main weakness is that it is not well demonstrated that this method outperforms existing ones. In particular, the trade-off in terms of rendering quality compared to the modest computational gain demonstrated in experiments.

As most reviewers said, this is more an evaluation of the current experimental setup, which is not complete enough, than of the method, and we think that a reworked version of this proposal - if it yields more convincing results - could be a very good contribution at another venue.